# Evidence of Constrained Divergence and Conservatism in Climatic Niches of the Temperate Maples (*Acer* L.)

Jake J. Grossman [1,2]

1   Biology Department, Swarthmore College, 500 College Ave, Swarthmore, PA 19081, USA;
    jgrossm1@swarthmore.edu
2   Weld Hill Research Building, Arnold Arboretum of Harvard University, 1300 Centre St,
    Boston, MA 02131, USA

**Abstract:** *Research highlights:* The availability of global distribution data and new, fossil-calibrated phylogenies has made it possible to compare the climatic niches of the temperate maple (*Acer* L.) taxa and assess phylogenetic and continental patterns in niche overlap. *Background and Objectives:* The maples have radiated from East Asia into two other temperate continental bioregions, North America and Eurasia (Europe and West Asia), over a roughly 60-million-year period. During this time, the Earth's climate experienced pronounced cooling and drying, culminating in cyclic periods of widespread temperate glaciation in the Pliocene to Pleistocene. The objective of this study is to use newly available data to model the climatic niches of 60% of the temperate maples and assess patterns of niche divergence, constraint, and conservatism in the genus's radiation out of East Asia. *Materials and Methods:* I assembled global occurrence data and associated climatic information for 71 maple taxa, including all species endemic to temperate North America and Eurasia and their closely related East Asian congeners. I constructed Maxent niche models for all taxa and compared the climatic niches of 184 taxa pairs and assessed phylogenetic signal in key niche axes for each taxon and in niche overlap at the continental and global scale. *Results:* Maxent models define a fundamental climatic niche for temperate maples and suggest that drought-intolerant taxa have been lost from the Eurasian maple flora, with little continental difference in temperature optima or breadth. Niche axes and niche overlap show minimal evidence of phylogenetic signal, suggesting adaptive evolution. Pairwise niche comparisons reveal infrequent niche overlap continentally and globally, even among sister pairs, with few taxa pairs sharing ecological niche space, providing evidence for constrained divergence within the genus's fundamental climatic niche. Evidence of niche conservatism is limited to three somewhat geographically isolated regions of high maple diversity (western North America, the Caucasus, and Japan). *Conclusions:* Over 60 million years of hemispheric radiation on a cooling and drying planet, the maple genus experienced divergent, though constrained, climatic niche evolution. High climatic niche diversity across spatial and phylogenetic scales along with very limited niche overlap or conservatism suggests that the radiation of the genus has largely been one of adaptive diversification.

**Keywords:** continental disjuncts; environmental niche modeling; Maxent; phylogenetic niche conservatism; paleobotany; phylogenetic signal; sister species; species distribution models





## 1. Introduction

The forests of the contemporary Northern Hemisphere would be unrecognizable in the absence of maples (genus *Acer* L.). After oaks (*Quercus* L. spp.), the maples comprise the second-most speciose tree genus in the temperate North [1]. Constituent species range from cosmopolitan generalists with large ranges (*A. pseudoplatanus*, *A. tataricum*) to endangered hyper-endemics (*A. pycnanthum* and *A. yangbiense* [2]). Yet, maple diversity is distributed unequally across the three bioregions in which these trees are found: upwards of 85% of maple species (of a total ~158 species [2,3]) are confined to East Asia (China, Japan, Korea,

Russia, the Indian Subcontinent, and Southeast Asia) with far fewer indigenous to the forests and woodlands of North America (nine species) and Europe and West Asia (Eurasia; 12 species). Each of these bioregional maple floras consist of continentally radiating clades giving rise to regionally sympatric, sister taxa (*sensu* [4,5]) as well as taxa for which the most closely related, extant sister taxon is regionally or continentally disjunct [6]. Yet, the significance of this variation in biogeographic histories for the evolution of the maples' ecological niches—the environmental conditions under which a particular taxon can persist (*sensu* Hutchinson [7])—remains unexplored. Only recently, has publicly available global distribution data, alongside genus-wide, fossil-calibrated molecular phylogenies, made it possible to determine how biogeographic history affects partitioning of ecological space.

Paleontological scholarship over the last 50 years and more recent phylogenetic reconstructions have provided a rough outline of the maple genus's origins and evolution. Boulter and colleagues [8] synthesized then-available fossil evidence and conclude that the maples diverged from their close relatives in Sapindaceae (e.g., *Dipteronia* and *Aesculus*) during the warm, wet early Eocene epoch (~60 mya) in the high latitudes of the Pacific Rim. They note, though, that ambiguities in the fossil record did not allow them to determine whether the genus originated in (what is now) East Asia or North America. Regardless, by the advent of the Oligocene (34 mya), a period of climatic drying and cooling, the maple genus had diversified considerably and radiated across the Northern hemisphere. Fossilized maple organs have been found in abundance in East Asia [9], North America [10], and Europe ([11], reviewed in [8,12]). Strikingly, fossils from some regions indicate considerable loss of maple diversity over the last ~30 million years of climatic cooling and drying; Wolfe and Tanai [10], for instance, document fossils of 91 maple species in 28 sections in western North America alone, compared to the five species (from five sections) extant in this region today.

Phylogenetic analysis of the extant maples has complemented fossil evidence, providing support for the "Out of Asia" theory of maple biogeography [6,13–16], in which the genus originated in East Asia and then radiated throughout the Northern Hemisphere. Following earlier molecular studies [1,17,18], two groups have recently released comprehensive phylogenies of *Acer*. Li and colleagues [14] present findings from a nuclear phylogeny, while Areces-Berazain and colleagues [15] synthesize both nuclear and plastome sequences. Though some differences in tree topology emerge across the two groups' work, both find evidence of an East Asian origin for *Acer,* divergence from sister genus *Dipteronia* Oliv. some 60–63 mya, and the coherence of most of the 13–16 historically delineated sections. Furthermore, in both cases, major sectional divergence within the genus appears to have taken place throughout the Eocene (34–56 mya), with subsequent speciation within sections occurring by the of the Miocene (23–5.3 mya), concomitant with multiple radiations into North American and Eurasia. I argue that, taken together, paleontological and phylogenetic lines of evidence suggest that initially widespread, then very limited, dispersal of the genus followed by vicariance, allopatric speciation, and ongoing, climate change-induced extinction produced the current diversity and distribution of the maples.

The contemporary North American and Eurasian maple floras resulting from this Northern Hemispheric radiation differ in their taxonomic and phylogenetic diversity. The existing temperate North American maple flora consists of nine recognized species from eight monophyletic sections (Figure 1), the remnants of a much larger maple diversity [10] whittled down by climate change-induced extinction and frequent isolation from the center of maple diversity in East Asia [16]. In their radiation across temperate North America the maple genus has colonized three distinct geographic regions: eastern North America (bounded by the Great Plains and the Atlantic Ocean), the West Coast (bounded by the Cascade Range and the Pacific Ocean), and the Interior West (ranging from Idaho and Wyoming, USA to northern Mexico). Subsequently, two clades have speciated further within North America: section Rubra (giving rise to *A. rubrum* and *A. saccharinum* [17]) and section Acer (the *A. saccharum* subspecies and *A. grandidentatum* [18]). All other maple sections extant in North America are represented by a single species.

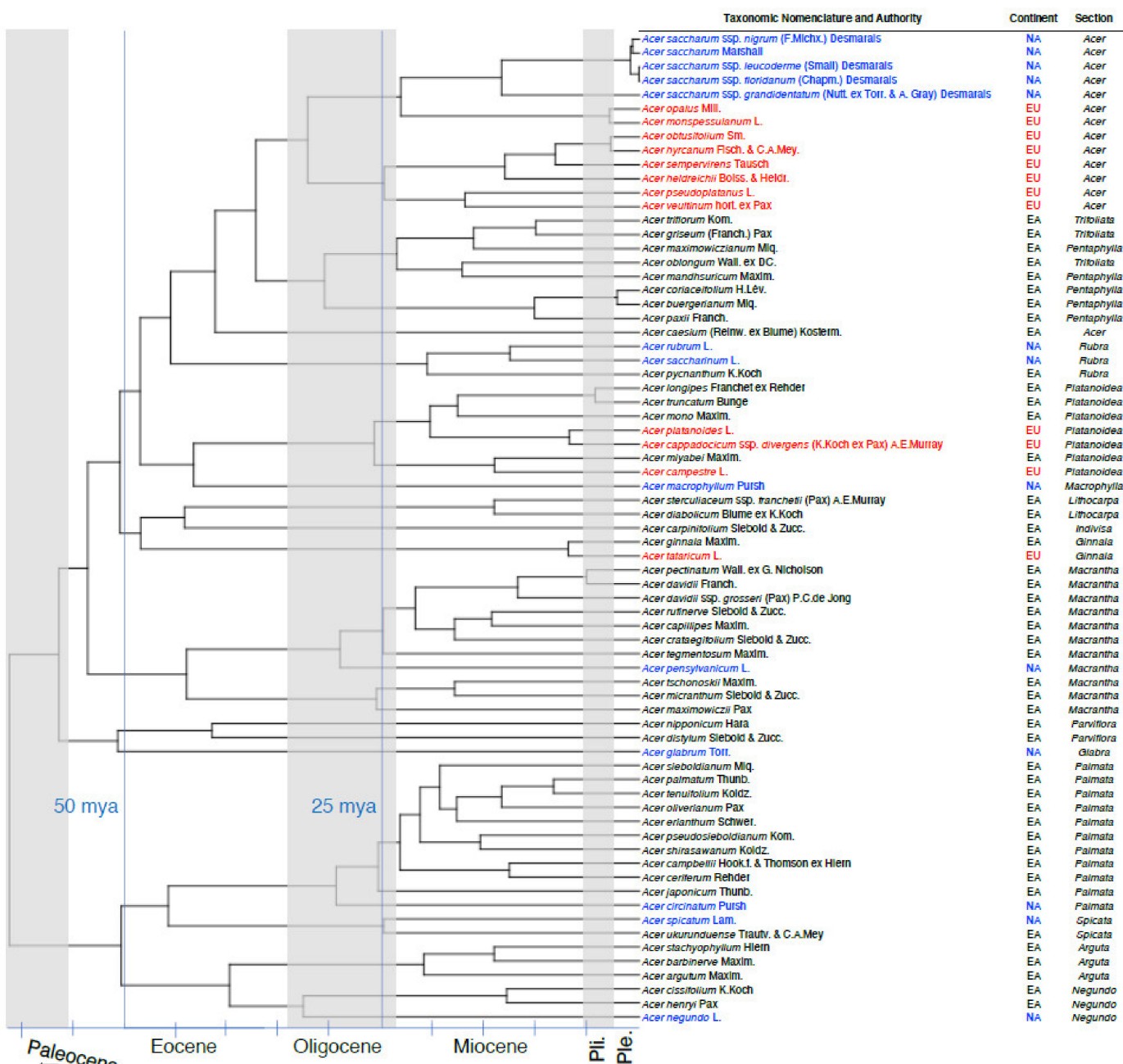

**Figure 1.** Study taxa in the maple (*Acer* L.) genus. Phylogenetic relationships, sectional classifications, and fossil-calibrated estimated divergence times for 71 taxa are redrawn from Areces-Berazain and colleagues' [15] supermatrix (nuclear and plastome) phylogeny. Species are also classified by bioregion: East Asia (EA, black), North America (NA, blue), or Europe and West Asia (EU, red).

In contrast, the extant Eurasian maple flora consists of 12 species from only three sections (Figure 1). Indeed, from a phylogenetic perspective, the Eurasian maple flora is even less diverse than that of North America [14,15], despite physical connection to and intermittent historical connectivity with the speciose maple flora of East Asia [16]. Though fossil evidence is not as synoptic as that available for North America and East Asia, available reports [12,19–22] indicate that Eurasia has also experienced a loss of maple diversity with post-Eocene cooling, drying, and successive glacial cycles. Of the 12 extant and well-defined Eurasian species considered here, eight correspond to the Acer clade, which is presently restricted to Eurasia and North America [18]; the *A. saccharum-grandidentatum* complex, its sole American constituent, is sister to all the Eurasian species [14,15]. An

additional three Eurasian species correspond to the Platanoidea group, which includes both East Asian and Eurasian species. Finally, the *A. tataricum* complex, an unusually cosmopolitan lineage and the only extant one in section Ginnala, extends across Asia from Korea and China to the Balkans and western Russia, with *A. tataricum* ssp. *tataricum* rounding out the temperate Eurasian maple flora [14,15]. Taken together, the Eurasian maples likely results from diversification of section Acer following a vicariant separation from East Asian sister lineages, co-occurring with multiple similar radiations involving the Platanoidea maples and range expansion of cosmopolitan *A. tataricum* across central Asia [15]. This triple radiation of the maples into Eurasia stands in contrast to the less speciose, but more phylogenetically diverse (eight-section) North American maple flora, as discussed above.

Over the course of their radiation into North America and Eurasia, the maples likely responded to post-Eocene global climatic cooling and drying interspersed with warmer, wetter cycles as did other high-latitude, temperate taxa: through a combination of adaptation, migration to more southerly refugia, and extinction [16,22–24]. Yet, such cycles of adaptation, migration, and extinction in response to cold, dry conditions followed by range expansion during milder periods likely did not affect all northern hemisphere floras equally [25–27]. Recent analyses have suggested that many lineages, including maples, experienced their highest losses of diversity in Europe, where conditions were at times extremely arid and cold. Meanwhile, diversity was protected in physiographically complex East Asia, where more extensive refugia were available, with North American floras responding more similarly to those of Europe [26]. More recent glacial cycles in the Quaternary (2.6 mya to present) have continued to shape the northern hemisphere temperate flora along these lines [25,27–29], leading to diversification of, for instance, the North American sugar maple complex [15,30]. This 60 million-year, pan-hemispheric radiation of *Acer* has resulted in considerable variation in the genus's morphology [31,32] and physiology [33]. Yet, it is not known whether the same can be said for the maples' ecological niche.

Interpretation of existing patterns in niche diversity at large spatial (e.g., hemispheric) and temporal (e.g., post-Eocene) scales is far from straightforward [34]. It has long been recognized that, all else being equal, recently diverged taxa are expected to share functionally significant traits, including those that structure their realized ecological niche [35]. Over evolutionary time, this similarity ought to deteriorate due to genetic drift and weak, fluctuating selection, producing a proportional, negative relationship between phylogenetic distance and ecological similarity, as in Brownian Motion models of trait evolution [36,37]. This pattern, of phylogenetic signal [38], provides a null expectation—albeit one that is rarely met—of what might be observed in the absence of pronounced stabilizing or directional selection on niche axes. Deviations for this pattern, including the absence of a relationship between phylogenetic distance and niche overlap or greater dis(similarity) between the two variables than would be expected based on Brownian Motion, cannot be interpreted in isolation from other evidence [34,39]. In the case of phylogenetic niche conservatism (PNC), the tendency of related taxa to retain key ecological characteristics over time [40], active debate has yet to fully resolve what constitutes evidence of PNC, whether PNC is an ecological pattern or process, and how PNC can be tested for given the availability of a credible phylogeny and distribution data for constituent taxa [37,38,41]. Given this, the following interpretation of phylogenetic patterns in the niche diversity of the maples follows Pyron and colleagues' [39] framework. This is to say that I interpret deviations in niche from the null (Brownian Motion or proportional) model of phylogenetic signal in light of the spatial and climatic diversity of the maples' Northern hemispheric distribution.

In this synthesis of the phylogenetic and biogeographic evidence of maple niche evolution, three potential patterns can arise suggesting deviation from the null expectation of a proportional relationship between phylogenetic distance and niche similarity: niche (1) divergence, (2) constraint, and (3) conservatism (*sensu* [39]). I discuss expected evidence for each of these patterns here in terms of inter- and intracontinental spatial scale. First, particular lineages within the genus may have experienced ecologically consequential

divergence as a result of evolution in novel conditions. This divergence may have occurred in response to both climate change and allopatric speciation over the genus's 60-million-year hemispheric radiation. In this case, continental maple floras should have distinct niches, reflecting climatic differences among East Asia, North America, and Eurasia [41,42]. Additionally, within continental floras, parapatric speciation is expected to have generated congeners showing low niche similarity, reflecting partitioning of both geographical and ecological space. Further evidence for divergence could also include a low level of niche overlap among sister taxa, which I treat here [4,5] as two taxa more closely related to each other than to any congener, inclusive of well-defined sub-species, such as those of the American sugar maple (*A. saccharum*).

Second, a pattern of constrained niche diversity may have arisen when species responded to novel conditions through development of distinct niches, all while bounded by lineage-level limits on their evolution [42,43]. In this case, continental floras are expected to occupy similar climates, reflecting a fundamental niche for the genus. Within continents, sympatric taxa are expected to have distinct climatic niches, reflecting a tendency of even sympatric taxa to partition ecological space [25,44,45].

Third and finally, Pyron and colleagues' [41] conceptualization of niche conservatism as biogeographic pattern (as distinct from PNC as process) would predict high levels of overlap both continentally and among sympatric species within continental floras. Particularly strong evidence for this pattern would emerge if more distantly related species (e.g., those from different sections) showed high niche overlap following radiation into a sympatric environment.

To date, though, such systematic, comparative analysis of the diversity in and phylogenetic context of the climatic niche occupied by the maple genus lags behind that executed for other lineages (e.g., [44–47]). Instead, rigorous modeling of maples' ecological niches has heretofore been restricted to ad hoc investigations of particular species of interest, usually with a focus on the climate change vulnerability of common forest species [48–53]. The persistent gap in our understanding of biogeographic patterns in and evolution of the ecological niche of *Acer* has motivated my exploration of the following questions:

1.  What are the climatic niches occupied by:

    a.  Particular taxa within the maple genus and
    b.  The continental maple floras of temperate East Asia, North America, and Eurasia?

2.  To what extent do the climatic niches of North American and Eurasian maple flora display evidence of (*sensu* [39]):

    a.  Weak selection and drift, such that niche axes and niche overlap vary proportionally with phylogenetic distance as predicted by a BM model (e.g., phylogenetic signal);
    b.  Divergence, such that continental niches are distinct and sympatric taxa and/or sister taxa show low niche overlap;
    c.  Constraint, such that continental niches are similar and sympatric taxa show low niche overlap; and/or
    d.  Conservatism, such that continental niches are similar and sympatric taxa— including those from different sections—show high niche overlap?

To address these questions, I assembled maple species occurrence records from publicly available data sources for 71 maple taxa (species or well-defined subspecies) spanning the genus's phylogeny and temperate distribution (Figure 1). After associating these occurrence points with relevant climatic data, I created and validated environmental niche models (ENMs) for each of these taxa and used a resampling method to make pairwise comparisons of niche similarity and niche optima and breadth between species pairs. Given the availability of presence data but not true absence data for the study taxa, I elected to use the Maximum Entropy or *Maxent* modeling [54,55] to build ENMs for each taxon. Maxent is widely used in ecological modeling; though it is vulnerable to sampling bias, misspecification of environmental background, inclusion of correlated predictor features, and other

pitfalls of ENM, these potential problems can be addressed through appropriate model specification [56]. Briefly, Maxent models integrate presence data for a taxon of interest and user-provided environmental data for each point throughout a user-defined spatial extent to produce estimates of the likelihood of the taxon's occurrence across that extent.

I interpret findings from these models in light of available fossil evidence and a genome-wide, fossil-calibrated phylogeny [15] to assess local to global biogeographic patterns in the ecological niche of *Acer*. Beyond providing a synoptic accounting of the climatic niches of maple taxa for which sufficient phylogenetic and distribution data are available, this approach may also shed light on the climatic forces affecting the distribution of rare and understudied taxa, including the 35 threatened maple species not included in the present study [2].

## 2. Methods

All analysis was carried out in the R statistical computing environment (Ver. 3.6.3, [57]). I manipulated spatial data using the "rgdal" package for implementation of the geospatial data abstraction library (GDAL [58]), the "sp" package [59], and the "maptools" package [60].

### 2.1. Distribution Data

I collected publicly available geographic occurrence data for 71 maple taxa from GBIF: the Global Biodiversity Information Facility ([61]; custom data export, datasets available at [62] Figure 1; Table S1 in Supplementary Materials). This coverage reflects: half of all recognized maple taxa; all recognized maple taxa in North America, Europe, and west Asia; all East Asian sister lineages of North American and Eurasian maples for which data were available; all temperate maple taxa from the Areces-Berazain et al. [15] phylogeny for which sufficient georeferenced occurrence data were available. This set of taxa includes representatives of all monophyletic sections and series except the monotypic section Wardiana and all ecologically widespread temperate maple taxa. For *A. pycnanthum*, which lacked sufficient GBIF coverage, I also collected geographic data from other published sources (Table S2). I excluded from analysis fully (sub)tropical maple taxa (e.g., *A. caudatifolium*, *A. laurinum*); Pleistocene relictual populations in North America (e.g., *A. skutchii*); taxa for which insufficient georeferenced presence data were available (e.g., *A. yangbiense*); and those for which a lack of systematic resolution made it challenging assign presence data to the right taxon with confidence (e.g., *A. mono* var. *mayrii*, *A. lobelii*). Additionally, of the 50 maple species identified as critically threatened, nearly threatened, or deficient in observations necessary to assess conservation status, only three (*A. griseum*, *A. miyabei*, and *A. pycnanthum*) are included [2]. These decisions and taxonomic treatments are unpacked further in Appendix A. Taxa selection thus constitutes a representative sampling of maple taxa with respect to the genus's phylogeny and temperate geographic distribution.

Following Hijmans and Elith [63], I cleaned all occurrence data as follows. With four exceptions, I removed all presence points outside of the native distribution of each taxon per van Gelderen [64]. For four widespread European species (*A. campestre*, *A. monspessulanum*, *A. platanoides*, and *A. pseudoplatanus*), I kept distribution data in western and northern Europe corresponding to the long-standing, naturalized range of these species [65,66]. I also removed all points of unknown or fossil provenance, those lacking latitude and longitude coordinates, and those with latitude and longitude coordinates identical to those of a conspecific sample. Sampling intensity varied widely such that the number of occurrence datapoints available for a given taxon ranged over five orders of magnitude and some regions (e.g., western Europe, Japan) were blanketed by geographically close presence points for particular taxa. To improve model comparability and avoid biasing models toward data from particular regions, I rarefied presence data to one point per 25 km using the *thin* command in the "spThin" package [67]. This procedure randomly removed data points to create a spatially thinned dataset without biasing the distribution of presence points for either widespread or endemic species (Figure S1). Cleaned and

thinned occurrence data for each taxon were then associated with climate data used for niche modeling.

## 2.2. Climate Data

I used the *getData* ("raster" [68]) command to automatically download climate data from the Worldclim database [69]. Climate data were gridded at a 2.5° resolution (1° = 111 km at the equator) and included four Bioclim variables: mean annual temperature (bio1, °C), temperature seasonality (bio4, °C), mean annual precipitation (bio12, mm/year), and precipitation seasonality (bio15, dimensionless). Though elevation is known to be an important factor in determining maple species' distributions [64], it was excluded from this study due to its frequent correlation with temperature and precipitation variables. Climatic data were not assessed at a finer spatial grain in order to provide niche estimates of sufficient generality to compare among taxa across a global scale. These four variables were then associated with each cleaned and thinned presence point.

I restricted climatic modeling of niche inputs to these four variables out of a desire to construct for each maple taxon an interpretable niche model that could be meaningfully compared with the model for another taxon from a different region or continent. For this reason, I chose variables associated with temperature and precipitation, which are known to constrain plant distribution across spatial scales [70,71], and did not include, for instance, the edaphic and hydrologic variables that might be relevant to one maple species and irrelevant to another. Furthermore, the four variables in question are generally not collinear across the maple genus's range (average collinearity following [72]: $\mu = 0.381$, $\sigma = 0.105$, range = [0.173, 0.615]; Table S3), making them suitable for inclusion in niche modeling. Other bioclimatic variables pertaining to temperature or precipitation (e.g., isothermality, bio3) could have been included, but these were largely correlated with the chosen variables across the study system and are less obviously interpretable to lay audiences and managers. Finally, temperature and precipitation are expected to vary in predictable ways at regional scales given prevailing climate change, and so are informative not only as determinants of existing species distributions, but also as potential indicators of climate change vulnerability [51].

## 2.3. Environmental Niche Modeling (ENM)

For each taxon of interest, I used the "dismo" package to build a Maxent ENM as follows using the taxon's cleaned, thinned presence points and associated climate data [73]. First, I determined an extent (Table S3) for each taxon as follows: I began with a rectangle including all presence and absence points and then expanded it to include terrestrial surfaces that might be hospitable to the taxon based on available information about its distribution [2,64–66]. For instance, ENMs of western North American species are bounded longitudinally by the Pacific Ocean and the Rocky Mountains. Within that extent, I then used the *randomPoints* function to generate a set of random background points equal in number to the given taxon's thinned presence points [73]. These background points were then also associated with climate data in the same manner as the presence points. I then generated a maxent model using all presence and background points (*maxent*) and default settings (allowing response classes to be set automatically) and built from this a prediction raster (*predict*) showing the likelihood of the given taxon's occurrence across the extent. From the Maxent model, I also extracted response plots (*response*) showing the effect of each climatic variable on the likelihood of the taxon's occurrence when each other variable was held at its median value. Finally, I evaluated variables' importance to the final model by retaining the permutation importance, determined by randomly permuting and resampling the importance of a given variable among presence and absence points, for each variable (Table S3, Figure S2 [72–74]).

I validated the Maxent ENM for each taxon using a cross-fold approach [63,75]. Ten times, I used 90% of the taxon's presence data to train a maxent model, with an extent and randomly computed background points generated as above. I then tested each model with

the withheld 10% of presence points. This allowed for the calculation of false (FPR) and true positive rates (TPR) and plotting of a TPR vs. FPR curve for testing each model using the *evaluate* function. The area under the curve (testing AUC) for such validation models is often used as an indicator of model performance; an AUC > 0.5 indicates a model that can differentiate between presences and absences better than expected by chance. Training AUCs are also presented for each taxon's Maxent model; the training AUC for a particular model can be interpreted as a metric of how well that model fits the training dataset (in this case, all thinned presence points). Yet, training AUCs should be interpreted with caution as they are often depressed for taxa with large or ecologically diverse distributions. As an alternative, I also calculated and report the True Skills Statistic (TSS; TPR + True Negative Rate [TNR]–1) alongside the training AUC. TSS ranges from −1 to 1, with values above zero indicating better model performance by chance. Minimum, medium, and maximum testing AUCs for the ten training models per taxon are reported alongside other model specifications and training AUCs and TSSs for each final model in Table S3. I also qualitatively validated the realism of output maps associated with my models for North American species against the maps generated by Ellenwood and colleague's [76] species distribution models.

*2.4. Niche Comparisons*

To assess similarity or divergence in ecological niche, I carried out three sets of pairwise niche comparisons as follows. I made pairwise 184 comparisons between all sympatric sister taxa (15 pairs), continentally (three pairs) or regionally (six pairs) disjunct sister taxa, all North American taxa (78 pairs), and European taxa (66 pairs). For context, I also compared select North American taxa with their closest (non-sister) extant Eurasian and East Asian co-sectional relatives (17 pairs) and compared select Eurasian taxa with their closest (non-sister) extant North American and East Asian co-sectional relatives (6 pairs). Comparisons are detailed in Table S4.

First, I used Warren and colleagues' [77] "ENMtools" package to execute a niche equivalency test and then a niche similarity (also called "background") test [78]. A niche equivalency test uses a boot-strapping approach to assess whether the niches of two taxa are statistically the same [79,80]. In this framework, Schoener's *D* is used as an index of niche similarity; a value of zero indicates two completely distinct niches and a value of one indicates two identical niches. For an identity test (*identity.test*), an empirical estimate of *D* is calculated based on the observed similarity of two species' niches—each described by a maxent ENM as detailed above. Presence points from both species are then pooled and resampled 100 times. For each resampling, new Maxent models are created and *D* is calculated. If the empirical *D* falls outside of this distribution of simulated, null *D* values, the niches in question are considered non-equivalent.

A similarity test (*background.test*) poses a related question: are two niches more or less similar than expected by chance? As Warren et al. [77] note, this question is especially important when comparing, for instance, disjunct sister taxa, which may occupy similar niches in different geographic spaces. Here, I employ a "symmetric" similarity test [77]. This test functions as does the identity test except that calculation of *D* is based on a comparison of background points from each taxon of interest. As such, the similarity test compares the environmental space that could possibly be occupied by individuals of each taxon [80]. Empirical values of *D* falling to the left or the right of the null distribution of bootstrapped *D* values indicate either a greater difference or similarity in niches than expected by chance [80].

Finally, following Molina-Henao and Hopkins [78], I used a permutation test to assess differences between each taxon pair in niche optima and breadths for two of the environmental variables used in the ENMs presented above: mean annual temperature and mean annual precipitation. For both variables, I treated the median value for all thinned presence points as the niche optimum and the difference between the 95th and 5th percentile as the niche breadth. I then merged presence points for both taxa and resampled

them randomly 100 times, each time keeping the number of points assigned to each taxon consistent with the actual number of presence points corresponding to that taxon. I then recalculated niche optima and breadths for each of these resampled datasets and compared empirical optima and breadths to the resulting null distributions. When empirical values fell outside the most extreme 5% of the null distribution, optima and/or breadths of the two taxa were considered distinct.

### 2.5. Phylogenetic Signal Analysis

I assessed whether particular axes of the maple climatic niche show evidence of phylogenetic signal and whether climatic niche overlap is associated with phylogenetic distance.

First, I tested for genus-wide phylogenetic signal (Blomberg's $K$ [38]) and local phylogenetic autocorrelation (local Moran's $I$ [81]) in the optimum and breadth of mean annual temperature and precipitation across Areces-Berazain and colleagues' [15] phylogeny (as in Grossman et al. [82]). To do so, I used the "phylosignal" [83] and "phylobase" [84] packages to map trait optima and breadths onto the maple phylogeny (*phylo4d*), calculate $K$ across the genus and $I$ for each tip, and carry out permutation tests to assess significance of general or local phylogenetic signal ($\alpha < 0.1$; *phylosignal* and *lipaMoran*).

Second, I used a Mantel test to assess whether niche overlap (empirical $D$) for each species pair was associated with phylogenetic distance [85,86] as estimated in the Areces-Berazain et al. [15] tree (*mantel* in the "vegan" package [87]). I conducted this analysis once for the North American maple flora ($N = 78$) and once for the Eurasian flora ($N = 66$), each time specifying 999 permutations. I did not apply a mantel test to compare the phylogenetic distance and niche overlap across all study taxa because pairwise niche overlap estimates for all pairs were not feasible or ecologically informative.

Finally, for a more liberal assessment of the dependence of niche overlap on phylogenetic distance, I explored the relationship between empirical $D$ and time since divergence across all three continental realms [88,89] using non-parametric rank correlation tests. I tested this relationship once including all pairwise comparisons ($N = 184$) and then for regionally sympatric taxa pairs ($N = 82$) and regionally allopatric taxa pairs ($N = 102$).

Use of Areces-Berazain and colleagues' [15] phylogeny, which integrates signal from both nuclear and plastome sequences, provided qualitatively similar results to analyses carried out with Li and colleague's [14] nuclear phylogeny (data not shown). As reviewed by both groups, these phylogenies generate trees with similar topologies to each other and to *Acer* phylogenies published by other workers.

## 3. Results

### 3.1. Niches Vary Across Acer Taxa and Continental Floras

All ENMs performed well in validation. Training AUCs ($\mu = 0.720$, $\sigma = 0.035$, range = [0.646, 0.922]) and TSSs ($\mu = 0.538$, $\sigma = 0.86$, range = [0.342, 0.784]) were also acceptable (Table S3). Figure 2 features likelihood-of-occurrence maps, validation curves, and response plots for two cosectional exemplar taxa; model output for all 71 study taxa is provided in Table S3 and Figures S2 and S3. ENMs indicate that the climatic factors affecting maple distribution are taxon-dependent in *Acer*. Different taxa vary in the extent to which each of the four climate variables considered in this study (means of and temporal variability in temperature and precipitation) constrain their distribution. Across the species included in this study, optimal mean annual temperature ranges from 3.6 °C to 17.8 °C and optimal mean annual precipitation ranges from 535 mm to 2009 mm. Niche characteristics also vary with bioregion (Table 1, Table S4). Most notably, while taxa have relatively similar temperature optima and breadths across continental floras, large bioregional differences emerged in precipitation niche characteristics. In particular, East Asian taxa grow in a wide range of wet and dry environments, while Eurasian taxa, by comparison, rarely occupy wet habitats (>1000 mm of precipitation a year) with North American taxa intermediate.

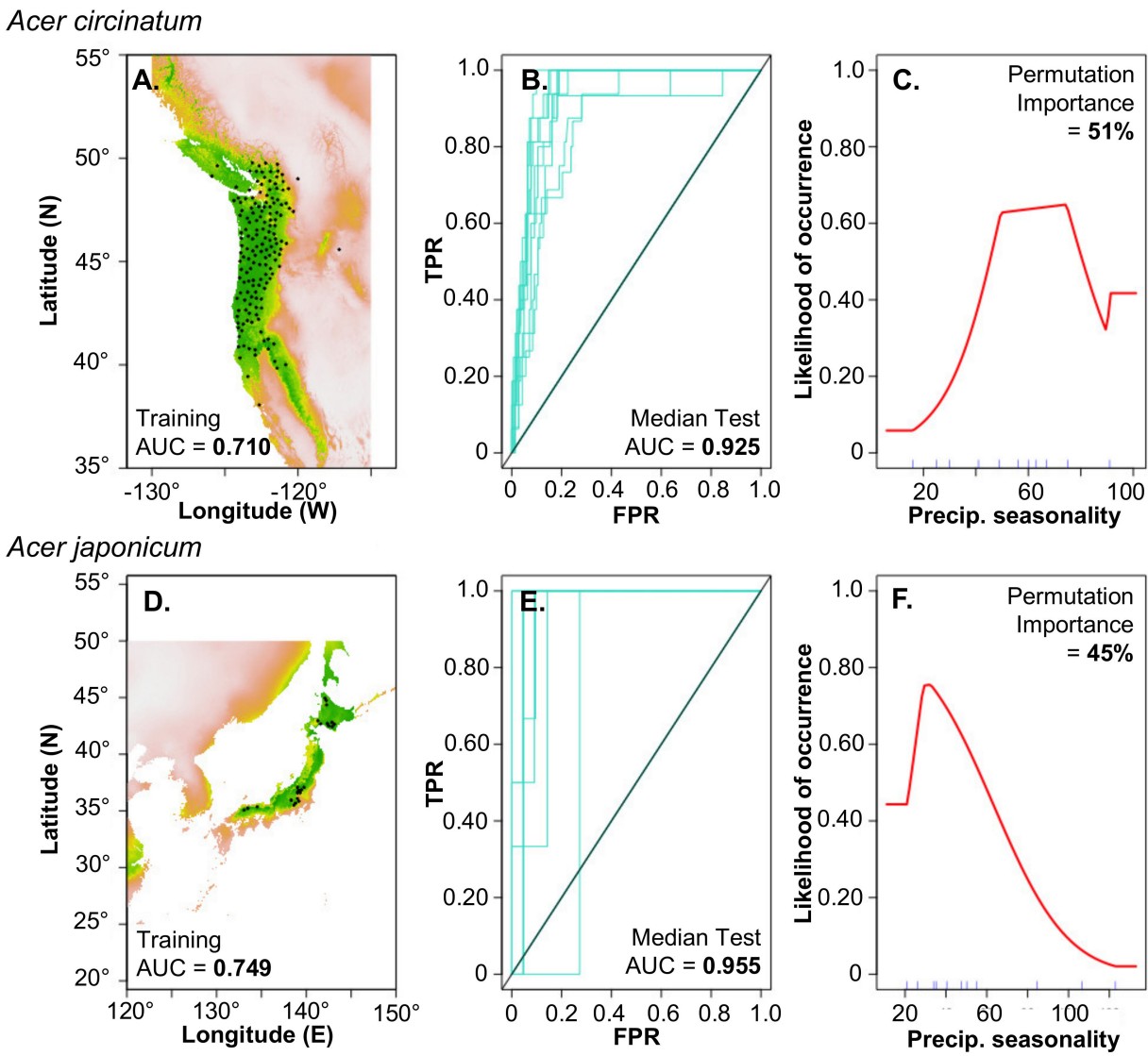

**Figure 2.** Maxent environmental niche model (ENM) output for two taxa in Section Palmata. (**A**,**D**) Prediction maps indicate locations with a high (green) to low (red/white) probability of occurrence. Actual, thinned occurrence points (black) and training area under the curve (AUC) are also provided. (**B**,**E**) True vs. false positive curves used for k-fold model validation along with median testing AUC (*N* = 10). (**C**,**F**) Precipitation seasonality (bio15, dimensionless coefficient of variation) was an especially important environmental variable predicting the distribution of both taxa. Curves indicate the likelihood of occurrence over a range of values of precipitation seasonality. Table S3 and Figures S2 and S3 give model validation data, response curves, and prediction maps, respectively, for all maple taxa.

**Table 1.** Minima and maxima of optima and breadths for two environmental axes of the niches of temperate maple taxa. See Table S5 for niche optima and breadths on a per-taxon basis.

| Continental Flora | Mean Annual Temp. (°C) | | Mean Annual Precip. (mm) | |
| :---: | :---: | :---: | :---: | :---: |
| | Optimum | Breadth | Optimum | Breadth |
| East Asia | (3.6, 17.8) | (4.3, 14.6) | (573, 2009) | (416, 2636) |
| North America | (4.9, 17.5) | (3.8, 16.6) | (535, 1437) | (369, 1956) |
| Eurasia | (4.8, 17.5) | (5.3, 10.6) | (545, 935) | (219, 907) |

### *3.2. Pairwise Niche Comparisons*

Outcomes from 184 pairwise comparisons of the climatic niches of maple taxa, including all possible pairwise comparisons in the North American and Eurasian bioregions, indicate a variable but generally low level of niche overlap (0.07 > *D* > 0.91, median = 0.29).

Only 15 taxa pairs had significantly overlapping climatic niches ($P_{\text{Identity}} \geq 0.05$, Table S4). Of these, four were allopatric pairs, each of them consisting of two non-sister taxa in the same section. The other 11 pairs were all sympatric: three distantly related pairs (each taxon was from a different section), seven sister pairs (of a total of 24 assessed), and one pair of closely related but not sister taxa. Only three of 78 North American pairs and five of 66 Eurasian pairs showed evidence of climatic niche overlap, suggesting a high degree of niche separation, even among closely related taxa in these bioregions. None of the three continentally disjunct sister pairs had overlapping niches, but all sister taxa pairs endemic to Japan did (Table 2).

Similarity tests for these pairs indicated no cases of greater niche similarity than expected, meaning that for no pair was empirically calculated niche overlap significantly larger than a bootstrapped distribution of null overlap values ($D > D_{\text{Null}}$ and $p_{\text{Similarity}} < 0.05$). As such, outcomes of similarity tests are given in Table S4 but are not discussed further here.

**Table 2.** Pairwise comparisons of climatic niches for sister taxa * in *Acer*. 24 sets of taxa are compared; all but three occur in the same continental bioregion. Epoch of divergence is drawn from Areces-Berazain et al. [15]). Pairs include 15 sympatric and 9 allopatric contrasts. Shading separates bioregions. Schoener's *D* is given for each pair, indicating the degree of overlap (0 = none, 1 = total overlap) in environmental niche as determined by Maxent modeling. *p*-values are given for an Identity Test and for permutation tests of similarity in Mean Annual Temperature (MAT) and Precipitation (MAP) optima and breadth. Non-significant *p*-values are bolded. Data corresponding to pairwise niche comparisons for non-sisters are given in Table S4.

| Sister Taxa | Bioregion | Epoch of Divergence | Sympatric? | Both Endemic to Japan? | Schoener's *D* | Identity Test | MAT Optimum | MAT Breadth | MAP Optimum | MAP Breadth |
|---|---|---|---|---|---|---|---|---|---|---|
| | | | | | | | | *p*-Value for Comparison (Significance Indicates Difference) | | |
| *A. rubrum* and *A. saccharinum* | N. America | Miocene | y | n | 0.63 | 0.01 | **0.21** | <0.001 | <0.001 | <0.001 |
| *A. saccharum* ssp. *saccharum* and *A. saccharum* spp. *nigrum* | N. America | Pleistocene | y | n | 0.65 | **0.05** | 0.01 | <0.001 | <0.001 | <0.001 |
| *A. saccharum* spp. *floridanum* and *A. saccharum* spp. *leucoderme* | N. America | Pleistocene | y | n | 0.66 | **0.15** | <0.001 | <0.001 | **0.77** | **0.81** |
| *A. pseudoplatanus* and *A. velutinum* | Eurasia | Miocene | y | n | 0.33 | 0.01 | <0.001 | **0.96** | **0.83** | **0.58** |
| *A. monspessulanum* and *A. opalus* | Eurasia | Pliocene | y | n | 0.44 | 0.01 | <0.001 | **0.42** | <0.001 | **0.86** |
| *A. hyrcanum* and *A. obtusifolium* | Eurasia | Pliocene | y | n | 0.38 | 0.01 | <0.001 | **0.68** | **0.32** | **0.05** |
| *A. cappadocicum* and *A. platanoides* | Eurasia | Miocene | y | n | 0.47 | 0.01 | <0.001 | **0.09** | <0.001 | **0.16** |
| *A. mandshuricum* and *A. oblongum* | E. Asia | Miocene | n | n | 0.32 | 0.01 | <0.001 | **0.29** | <0.001 | <0.001 |
| *A. sterculiaceum* ssp. *franchetii* and *A. diabolicum* | E. Asia | Miocene | n | n | 0.21 | 0.01 | <0.001 | **0.19** | <0.001 | **0.08** |
| *A. barbinerve* and *A. stachyophyllum* | E. Asia | Miocene | n | n | 0.23 | 0.01 | <0.001 | <0.001 | 0.02 | **0.45** |
| *A. griseum* and *A. triflorum* | E. Asia | Miocene | n | n | 0.23 | 0.01 | <0.001 | **0.35** | **0.13** | **0.70** |
| *A. micranthum* and *A. tschonoskii* | E. Asia | Miocene | y | y | 0.37 | **0.05** | <0.001 | 0.03 | 0.01 | **0.78** |
| *A. henryi* and *A. cissifolium* | E. Asia | Miocene | n | n | 0.43 | 0.01 | <0.001 | **0.86** | <0.001 | **0.50** |
| *A. campbellii* and *A. ceriferum* | E. Asia | Miocene | y | n | 0.45 | 0.01 | **0.30** | **0.12** | <0.001 | <0.001 |
| *A. buergerianum* and *A. coriaceifolium* | E. Asia | Pleistocene | y | n | 0.33 | 0.01 | <0.001 | **0.23** | **0.28** | 0.03 |
| *A. davidii* and *A. pectinatum* | E. Asia | Pliocene | y | n | 0.73 | **0.33** | <0.001 | **0.73** | <0.001 | **0.09** |
| *A. palmatum* and *A. tenuifolium* | E. Asia | Miocene | y | y | 0.44 | **0.06** | <0.001 | 0.02 | <0.001 | 0.04 |
| *A. pseudosieboldianum* and *A. shirasawanum* | E. Asia | Miocene | y | n | 0.29 | 0.01 | 0.02 | **0.82** | <0.001 | **0.7** |
| *A. longipes* and *A. truncatum* | E. Asia | Pliocene | y | n | 0.31 | 0.01 | <0.001 | <0.001 | <0.001 | 0.45 |
| *A. capillipes* and *A. rufinerve* | E. Asia | Miocene | y | n | 0.71 | **0.32** | **0.88** | **0.72** | **0.50** | **0.17** |
| *A. distylum* and *A. nipponicum* | E. Asia | Eocene | y | y | 0.91 | **0.80** | **0.57** | <0.001 | 0.04 | **0.34** |
| *A. spicatum* and *A. ukurunduense* | N. America/E. Asia | Oligocene | n | n | 0.27 | 0.01 | **0.89** | <0.001 | <0.001 | <0.001 |
| *A. campestre* and *A. miyabei* | Eurasia/E. Asia | Miocene | n | n | 0.25 | 0.01 | 0.01 | 0.01 | <0.001 | **0.6** |
| *A. tataricum* and *A. ginnala* | Eurasia/E. Asia | Miocene | n | n | 0.40 | 0.01 | <0.001 | <0.001 | <0.001 | <0.001 |

* sister taxa are defined as pairs of taxa more closely related to each other than to any congener included in the present study per Areces-Berazain and colleagues' (2021) phylogeny.

### 3.3. Phylogenetic Signal Analysis

Values for Blomberg's *K* were not significant ($p > 0.05$) for temperature and precipitation optima and breadths, providing no evidence of lineage-wide phylogenetic signal in these niche axes. Likewise, local phylogenetic autocorrelation was limited, with little patterning beyond small pockets of local phylogenetic signal for some niche axes in, for instance, sections Trifoliata-Pentaphylla and Palmata (Figure S4).

The outcome of Mantel tests assessing the relationship between phylogenetic distance and niche overlap differed between the North American and Eurasian temperate maple floras. There was no evidence of a phylogenetic signal in niche overlap among the North American maples ($r = 0.574$, $p = 0.337$), but a marginal signal for Eurasian taxa ($r = 0.194$, $p = 0.078$), suggesting that more closely related Eurasian taxa show a slightly lower degree of niche overlap.

Across all taxa included in this study, and without reference to continental bioregion, phylogenetic distance was also, via rank-correlation test, unrelated to niche overlap ($\rho = -0.086$, $p = 0.247$). Yet, comparison of pairs living in regional sympatry vs. allopatric pairs revealed a hidden phylogenetic pattern. Though niche overlap of allopatric taxa showed no phylogenetic signal ($\rho = -0.074$, $p = 0.510$), there was a significant negative relationship between phylogenetic distance and niche overlap for sympatric taxa pairs ($\rho = -0.282$, $p = 0.004$), suggesting at least some adaptive partitioning of the climatic niche by co-occurring taxa (Figure 3).

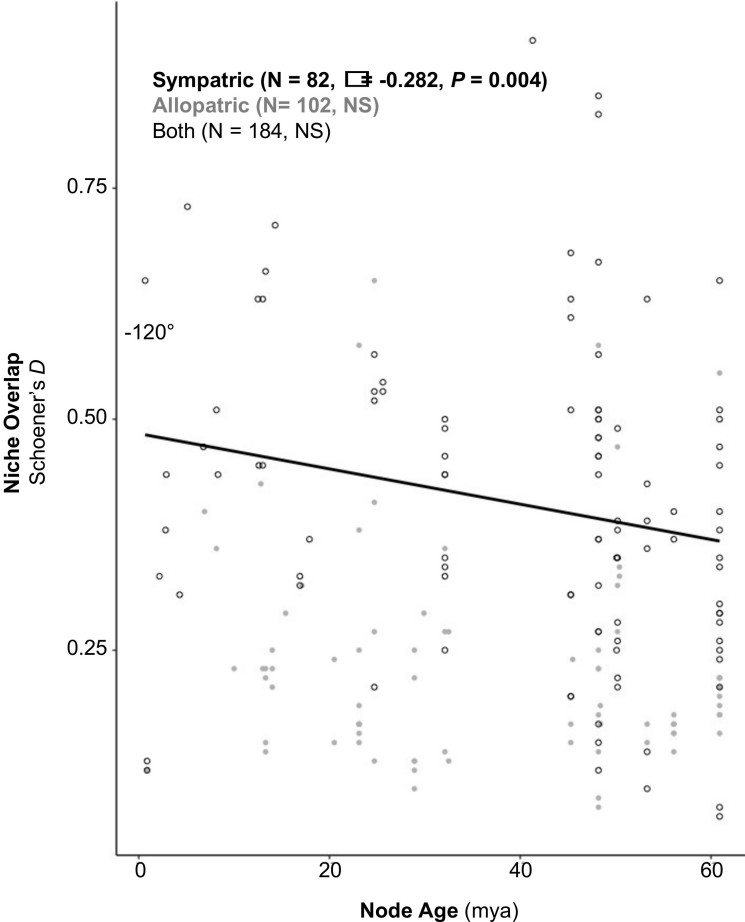

**Figure 3.** Node age (mya) and degree of climatic niche overlap (*D*, in which *D* = 1 indicates total overlap and *D* = 0 indicates no overlap) were uncorrelated for all taxa pairs considered and for allopatric pairs (gray, closed circles). However, recently diverging sympatric taxa pairs (black open circles) had significantly more similar climatic niches than did more distantly diverging taxa (simple linear regression line shown, adjusted R$^2$ = 0.04). Full dataset can be accessed in Table S4.

## 4. Discussion

Through species distribution modeling and integration of phylogenetic and biogeographic data, I present here the most synoptic accounting to date of the ecological niche of the temperate *Acer*, focusing on niche evolution concomitant with the genus's radiation into North America and Eurasia. In this work, the maples' current distribution across the temperate North shows patterns of climatic niche divergence, constraint, and conservatism. Below, I integrate findings across the entire maple genus and for the maple floras of North America and Eurasia.

### 4.1. The Global Acer Climatic Niche

Through their hemispheric radiation, maple species' optimal temperature niches have attained their present breadth of 3.6 °C to 17.8 °C, covering two-thirds of the range that typically defines a temperate climate (−3.0 °C to 18.0 °C MAT; [90]; Table 1). Indeed, across the Northern Hemisphere, mean annual temperatures slightly above 0 °C and mean annual precipitation in excess of 500 mm/year seem likely to be the climatic constraints limiting maple colonization. Notably, maple floras on each continent occupy very similar ranges of mean annual temperature, suggesting a fundamental temperature niche, whereas drought-intolerant species appear to have been eliminated from the Eurasian flora and, to a lesser extent, the North American one (Table 1). These continental differences in precipitation niche are likely due to the prevalence of a more permissive (wetter and less frequently glaciated) landscape in East Asia than in either Eurasia or North America during the maples' post-Eocene diversification [16,22,23,26,27]. As such, at the continental (hemispheric) scale, the genus's niche shows evidence of both constraint/conservatism in temperature niche and divergence in precipitation niche [39,42]. I borrow Yang and colleague's [91] concept of "constrained divergence" to describe this pattern: in which trait divergence is limited by fundamental constraints to adaptive evolution. In the context of Pyron and colleague's framework, constrained divergence occurs when fundamental limitations on adaptation interact with divergent evolution in response to novel or changing climatic conditions.

In contrast, phylogenetic analysis of the maples' climatic niches leads to rejection of the null model of diversification through drift and weak selection. There was no evidence of phylogenetic signal in either the optima or breadths of the maples' mean annual temperature or precipitation niches, meaning that closely related taxa were no more or less likely than expected by chance to be similar in these niche axes [36,37]. Furthermore, pairwise niche overlap was not associated with phylogenetic distance in the North America maple flora and was slightly negatively correlated with overlap in Eurasia. Indeed, the one element of clade-wide phylogenetic signal that emerged is the exception that proves the rule. Phylogenetic distance and niche overlap were uncorrelated across the entire set of pairwise taxa comparisons, consistent with expectations for niche divergence, constraint, or conservatism [39]. Yet, when sympatric and allopatric taxa were assessed separately, this pattern only held for those whose ranges do not overlap. On the other hand, *sympatric* taxa showed a modest pattern of decreasing niche overlap with evolutionary distance (Figure 3). This finding suggests that only in the absence of diversifying adaptation to novel conditions could phylogenetic signal in niche overlap emerge.

Analysis of sister taxa has been widely employed as a means for assessing phylogenetic patterns in niche diversity across clades [5] and generally indicated niche divergence for *Acer*. Across 24 pairs of temperate maples, I found no relationship between niche overlap (*D*) and sister pairs' degree of sympatry, continent of origin, or phylogenetic distance (Table 2, Table S4). No Eurasian or continentally disjunct sister taxa had significantly overlapping niches, but two very recently diverged, sympatric pairs of North American *A. saccharum* subspecies and five pairs of sympatric East Asian taxa did. In general, then, comparison of the climatic niches of taxa more closely related to each other than to any other congener demonstrated widespread divergence; this was universally the case among allopatric pairs. To this point, all three sister pairs endemic to the Japanese archipelago showed significant and high niche overlap (Table 2), even in the case of *A. distylum* and

*A. nipponicum*, which likely diverged in the Eocene (~41.3 mya). It seems likely, then, that intermittent isolation (e.g., due to sea level rise) in a stratigraphically complex environment has facilitated conservatism of the maples' climatic niche during sympatric speciation in Japan [26]. Yet, outside of Japan, most sister taxa, especially those not belonging to the same species complex, have speciated allopatrically or in less restricted ecoregions producing a predominant pattern of niche divergence.

*4.2. Niche Divergence, Constraints, and Conservatism Vary Across Scales in the North American Maple Flora*

Within the North American maple flora, patterns of climatic niche diversity and overlap conform to expectations for three scenarios: divergence, constraint, and conservatism. Evidence for each of these patterns varies across spatial and phylogenetic scales.

At intercontinental and regional spatial scales, the North American maple taxa appear to have experienced niche divergence through their radiation into and across the continent. Of the 14 cases in which a North American taxon had a co-sectional relative in either Eurasia or East Asia, only two pairs, *A. grandidentatum-A. monspessulnaum* ($D = 0.58$, $P_{Identity} = 0.07$) and *A. pensylvanicum-A. grosseri* ($D = 0.29$, $P_{Identity} = 0.06$), showed significant niche overlap. Instead, most North American maples are highly distinct in their climatic niche from their closest relatives in East Asia or Eurasia. For example, Section Rubra includes only a single extant temperate East Asian species, the Japanese *A. pycnanthum*, which diverged from its North American co-sectional species 20.5 million years ago [92,93]. Though its two closest relatives, the sympatric North American *A. rubrum* and *A. saccharinum* (diverging 10–13 mya [14,15] occupy more similar niches to each other ($D = 0.65$) than to *A. pycnanthum* ($D = 0.14, 0.24$), all three species show significant niche differentiation ($P_{Identity} = 0.01$). Furthermore, the North American maple flora more generally exemplifies a pattern of divergence across bioregions within the continent: no pair of American taxa hailing from different regions has a shared niche ($P_{Identity} \leq 0.05$; Table S4). As such, the novel climatic conditions presented by radiation into North America and across regions within the continent appear to have facilitated adaptive diversification in the maples' climatic niches.

Particularly within eastern North America, though, patterns of niche conservatism emerge at small, infraspecific phylogenetic scales juxtaposed against and constrained by niche evolution at larger, sectional scales. This pattern is exemplified by the diversification of the *A. saccharum* subspecies and *A. grandidenatatum*, the sugar maples. Though typically treated as a single widespread species complex consisting of up to seven sub-species (five, including *A. grandidentatum*, are considered here), the sugar maples are likely in the process of both sympatric and parapatric speciation, as evidenced by diversity in genetics [14,15,30], morphology [64], and distribution (Figure S3). Indeed, my comparison of the climatic niches occupied by these sub-species offers support for Oterdoom's [94], p. 18) proposal that "the sugar maple (*Acer saccharum*) has evolved subspecies to fill different niches in the vast North American continent." Of the five taxa in the complex that I considered in this analysis, two pairs northeastern *A. saccharum* ssp. *saccharum* and ssp. *nigrum* and southeastern ssp. *floridanum* and *leucoderme*,—both of them highly sympatric and recently diverged sister taxa—show evidence of significant niche overlap ($D = 0.65$–$0.66$, $P_{Identity} \geq 0.05$) and similarity in niche axes (Table 2, Tables S4 and S5). Niche overlap between these sympatric pairs, suggests that, at a small phylogenetic scale (<1 million years of divergence), niche conservatism predominates. Yet, none of the *A. saccharum* subspecies studied here show significant niche overlap with allopatric *A. grandidentatum* (diverging 13.3 mya) or with the other five, more distantly related sympatric maple taxa native to eastern North America (*A. rubrum, A. saccharinum, A. negundo, A. spicatum,* and *A. pensylvanicum*). Nor do any these sympatric taxa from outside the sugar maple group overlap in their climatic niches with each other or the sugar maples (Table 2, Table S4). In this case of eastern North America, then, sympatry alongside the absence of niche overlap provides evidence of constrained niche evolution at phylogenetic scales above the species level. In this region, more distantly related maple taxa are likely confined by fundamental

climatic constraints to relatively wet environments [25] but have diversified over time within this ecological space to realize distinct climatic niches.

Finally, a single pair of maples confined to the North American West Coast share an overlapping niche, providing limited evidence of either niche conservatism or convergent evolution. Distantly related but sympatric *A. macrophyllum* and *A. circinatum* share a wide breadth in precipitation niche and have statistically indistinguishable climatic niches ($D = 0.65$, $P_{\text{Identity}} = 0.15$; Table S4). Diverging in the mid Paleocene (60.9 mya), these two maple taxa might either have sorted into habitats associated with a conserved maple climatic niche (niche conservatism) or converged in their niche following considerable niche divergence (convergent evolution [95]). Regardless, this pattern of sympatric niche overlap among distantly related taxa is the only one I documented in the North American maple flora.

### 4.3. Constrained Divergence in Most Eurasian Maple Niches Contrasts with Conservatism in the Caucasus

As documented for the North America flora, the Eurasian maples appear to have diverged in their niches during their radiation into and within the continent. Neither of two European maples (*A. campestre* and *A. tataricum* ssp. *tataricum*) with sister taxa in East Asia show niche overlap with their sisters, despite relatively recent divergence times in the mid- to late Miocene, providing evidence of adaptive divergence. Additionally, within Eurasia, only five of 66 possible taxa pairs overlap significantly in their climatic niche; four of these pairs are sympatric species endemic to the Caucasus region (discussed below). The last is a pair in section Acer, *A. sempervirens* and *A. velutinum,* which share similar niches ($D = 0.65$, $P_{\text{Identity}} = 0.17$) despite their allopatric distribution. Finally, none of the cases of significant climatic niche overlap include *A. tataricum* ssp. *tataricum*, a phylogenetic outlier only distantly related to and largely allopatric with the remainder of the Eurasian maple flora, suggesting a role for niche diversification in its arrival in Eurasia.

Yet, the maples' climatic niche diversity within Europe appears to have been constrained by periods of glaciation and climate change, which have historically limited the availability of cool, wet niche space [6,26,27]. As a result, the relatively phylogenetically uniform Eurasian maple flora occupies a narrow precipitation niche (Table 1), with most cosmopolitan and Mediterranean taxa occupying distinct niches despite considerable range overlap (Table S4). Typical of this pattern are the four cosmopolitan Eurasian maples—*A. campestre, A. monspessulanum, A. platanoides,* and *A. pseudoplatanus*. Though they have become naturalized across Europe and into West Asia, all show significant niche differentiation compared to one another and to more endemic, but co-occurring species. Additionally, interestingly, though few European maple taxa show differentiation in niche breadth compared to their sympatric congeners, differences in optimal temperature and precipitation niche are quite common, and frequently significant (Table S4). Taken together, this evidence suggests that the maples' climatic niche diversity in Eurasia has generally been shaped by sympatric taxa partitioning a constrained ecological space.

Though the Eurasian maple flora on the whole shows evidence of constrained niche divergence, there does appear to be some evidence of conservatism among those species from sections Acer and Platanoidea whose ranges include the Caucasus region, located between the Black and Caspian Seas. Niches of four pairs of these maples (Table 2) show a significant degree of similarity ($0.57 < D < 0.83$, $P_{\text{Identity}} > 0.05$). This pattern of niche overlap can be interpreted in light of the biogeographic history of the present-day Caucasus. From the early Oligocene (34 mya) through the late Miocene (6–6.5 mya), the contemporary Caucasus was formed through a series of orogenic plate collisions coincidental with periodic inundation due to changes in the global climate [96]. Consequently, during this time, the Caucasus region included, in turn, multiple isolated landmasses; a bridge between previously isolated European, Asian, and African bioregions; and, in the case of the Greater Caucasus mountains, a small island isolated by sea straits through the late Miocene [97]. Coinciding with the intra-sectional speciation of most extant *Acer* taxa, evolution during this complex geological history could have formed the backdrop for the development of

the region's current maple flora, in which multiple species from two sections have overlapping climatic niches. The relatively high maple diversity of this region was also likely supported by the availability of refugia during the Pleistocene glaciations [98]. Though it is possible that the high degree of niche overlap in the Caucasus has resulted from cross-sectional niche convergence during the region's post-Oligocene history, the evidence reviewed here suggests at least some role for sorting of maple taxa into a region aligned with their conserved climatic niche.

## 5. Conclusions

The work presented here includes a synoptic accounting of the well-characterized, temperate North American and Eurasian maple taxa and their East Asian nearest relatives. Coverage focuses on widespread species and those for which molecular [14,15] and reliable, representative distribution data are available. However, the findings from this analysis must be interpreted in light of several inherent limitations to my approach.

First, I did not assess the niches of all maple species. Some endemic, temperate East Asian taxa with small or poorly documented ranges are excluded, as are the tropical and subtropical maples, including relictual populations in Mesoamerica. As such, inference regarding the climatic niche characteristics of, for instance, the temperate East Asian continental flora should be interpreted with some caution. Yet, I contend that the analysis I present here is likely to be robust to addition of endemic East Asian species since it includes coverage across the genus's sections as well as temperate taxa occupying the coldest and driest habitats in which maples are found. Furthermore, consideration of endemics with historically small ranges that have been further reduced by habitat loss and relictual populations with historically larger ranges may distort modeling of their distributions by excluding what was once part of their realized niches [56,99].

Second, in selecting presence data for each maple taxon included in this study, I elected to strike an imperfect balance between two contrasting approaches to the problem of presence points outside of a taxon's documented native range. On one hand, it is well known that species distributions are generally not at equilibrium such that historical contingency, as well as deterministic climatic and non-climatic variables determine the "native" distribution of any given species at any given moment, leading to incompletely filled niches [100,101]. Species invasions and the global distribution of many horticultural varieties attest to this fact and provide justification for including presence points outside of species' native ranges in niche modeling [102]. Yet, inclusion of points associated with naturalized or intentionally planted trees outside of their native range may also introduce considerable bias into species distribution models. Many such presences occur in urbanized settings, in which human engineering of the environment frequently has altered and often homogenized local hydrology, temperature, soil conditions—all critical aspects of ecological niche for woody plants [103–105]. Furthermore, GBIF presence points from outside of species' native ranges often correspond to collections of botanic gardens, at which collected plants are nurtured through their more physiologically vulnerable (juvenile) stages [106–109]. Actual naturalization in these locations may be impossible, even if horticultural care allows for survival of transplanted adults. Taking all of these considerations into account, I chose to include the naturalized of four well-studied Eurasian species in my analysis, while otherwise excluding naturalized or planted occurrences of other taxa outside their native ranges. Exploration of niche models for widely naturalizing or invasive species (e.g., *A. platanoides*) may provide new insights into the realization of these species' fundamental climatic niches in a changing climate [75,101].

Finally, my implementation of Maxent involved several choices made to optimize some modeling objectives at the expense of others. Typically, species distribution modeling for a single taxon entails critical consideration of predictor variables that might be relevant to the distribution of the taxon in question (e.g., [52,75,78]). In the context of the present study, my interest was in selecting a common set of variables known to be widely relevant to the distribution of temperate woody plants and to use these to construct Maxent models

for all study taxa (e.g., [44]). In doing so, I have created a set of models that can be compared meaningfully among maple taxa. This approach allows for meaningful comparison of the role of, for instance, mean annual temperature, in structuring the distributions of section Rubra: *A. rubrum*, *A. saccharinum*, and *A. pycnanthum*. A key drawback to this approach is that it would likely be possible to create a *better* model—meaning a more predictive one—of the distribution of any particular taxon. Furthermore, despite consideration of the well-known drawbacks of doing so [56,110], I have followed the same rationale and allowed Maxent's default setting to determine automatically which feature classes are applied to each model. (These feature classes specify the allowable relationships between predictors and habitat suitability estimates. Models with more presence data are allowed to explore a wider array of fits.) Again, the implications of this course are that each particular model might not be the optimal one possible, but the approach applied to the development of each model was uniform.

Going forward, new molecular and occurrence data, including for species excluded from this analysis, will become available, allowing for extension of the modeling framework presented here to include a greater proportion of maple taxa. Future modeling approaches may also take into account niche axes excluded from this analysis, including edaphic factors [111], biotic interactions [112], and human impacts [44]. Finally, data assembled and curated for this project constitute an important resource for phylogenetically informed prediction of climate change impacts on the global maple flora [51] and conservation planning for taxa of concern [113].

**Supplementary Materials:** The following are available online at https://www.mdpi.com/article/10.3390/f12050535/s1, Figure S1: Example of random thinning procedure, Figure S2: Response plots for climatic niche models of each maple taxon, Figure S3: Climatic niche maps for each maple taxon, Figure S4: Phylogenetic signal in niche axes, Table S1: Longitude and latitude for all cleaned and thinned presence and background points, Table S2: All non-GBIF occurrence data points gathered for *A. pycanthum* [114–116], Table S3: Maxent climatic niche model extents and output for all taxa, Table S4: Results of pairwise comparisons among *Acer* taxa, and Table S5: Temperature and precipitation niche optima and breadths for each study taxon.

**Funding:** The author was funded by the Putnam Postdoctoral Fellowship of the Arnold Arboretum of Harvard University and a Swarthmore College Faculty Research Support Award.

**Institutional Review Board Statement:** Not applicable.

**Informed Consent Statement:** Not applicable.

**Data Availability Statement:** All cleaned/processed data are made available as supplemental materials associated with this article.

**Acknowledgments:** Analysis described in this paper was refined through conversations with Robin Hopkins, Antonio Serrato-Capuchina, Benjamin Goulet-Scott, and Charlie Hale. Jinhua Li and Joeri Strijk provided helpful access to their groups' phylogenies. Finally, the author wishes to thank Bradley Davidson and Samridhi Chaturvedi for helping to arrange computing resources. The author was supported in this work by a Katharine H. Putnam Fellowship in Plant Science at the Arnold Arboretum of Harvard University. Four anonymous reviewers provided suggestions that improved the quality of this manuscript.

**Conflicts of Interest:** The author declares no conflict of interest.

## Appendix A. Explanation of Selection and Taxonomic Treatment of Study Taxa

Taxa considered in this study include all of those detailed in Areces-Berazain [15] and colleagues' synoptic study of the genus with exceptions detailed as follows.

The focus of this work is the temperate maple flora of East Asia, North America, and Eurasia. For this reason, several taxa were excluded because they are adapted to a subtropical climate. East Asian taxa excluded based on this criterion are *A. laurinum* (section Rubra); *A. caudatifolium*, *A. rubescens*, and *A. morrisonense* (section Macrantha); the primary subtropical clade within section Palmata (*A. calcaratum* through *A. pubipalmatum* in Figure 3

of Areces-Berazain et al. [15], excluding *A. sieboldianum*); and *A. pubinerve* (section Palmata). These taxa were excluded not only to retain a focus on temperate taxa, but also because representative geographical data across their ranges were unavailable for many of them (*A. rubescens*, *A. calcaratum*, *A. olivaceum*, *A. elegantulum*, *A. pauciflorum*, *A. pubipalmatum*, and *A. pubinerve*). Similarly, Eurasian and North American populations were excluded if they were small, poorly characterized, and likely the result of post-Pleistocene range contraction. Populations or taxa excluded for this reason are Mesoamerican *A. skutchii* and *A. binzayedii* (section Acer; *A. binzayedii* was not included in Areces-Berazain and colleagues' work) and relictual populations of some Eurasian taxa in North Africa and the Middle East (for which there is also limited information about distributions).

I also did not include several taxa included in Areces-Berazain and colleagues' [15] study for which there was not sufficient publicly accessible information regarding a taxon's distribution across its known or supposed range. These taxa are *A. obtusatum* and *granatense* (Section Acer, considered ssp. of *A. opalus* in many treatments and so lacking independent distribution information); *A. discolor* (grouped with *A. coriaceifolium* in this study) and *A. pentaphyllum* (Trifoliata-Pentaphylla clade); *A. yangbiense* and *A. pilosum* (grouped with *A. caesium* in this study); *A. sikkimense* (section Macrantha); *A. amplum* ssp. *catalpifolium*, *A. maiotense*, *A. amplum ssp. amplum*, *A. lobelii* (considered a ssp. of *A. platanus* in many treatments and so lacking independent distribution information), *A. shenkanense*, and *A. cappadocicum* spp. *sinicum* (Section Platanoidea); *A. tsinglingense* (section Lithocarpa), *A. tataricum* ssp. *aidzuense* and spp. *semenovii* (section Ginnala, for which there is not sufficient distribution data separating these taxa from the other *A. tataricum* subspecies); *A. pseudoseiboldianum* var. *microsieboldianum, A. laevigatum,* and *A. takesimense* (Section Palmata); and *A. acuminatum* and *A. stachyophyllum* spp. *betulifolium* (Section Arguta).

Finally, the systematics of some putative species complexes are still in the process of being resolved. As a result, for a few of these cases, I exclude from analysis a few taxa currently named as subspecies/varieties but apparently, per Li et al. [16] and Areces-Berazain et al. [15], deserving of reclassification. These are *A. mono* var. *mayrii* (section Platanoidea; *A. mono* ssp. *mono* is included) and *A. sterculiaceum* ssp. *sterculiaceum* (Section Lithocarpa; *A. sterculiaceum* ssp. *franchetii* is included). Pending further resolution, I also allow Areces-Berazain and colleagues' *A. cappadocicum* ssp. *divergens* to stand in for the more widely distributed and sympatric European *A. cappadocicum* (ssp. *cappadocicum*). Similarly, I treat their *A. trautvetteri* as synonymous for both *A. heldreichii* var. *heldreichii* and var. *trautvetteri*.

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
