# Peer review of "Evidence of Constrained Divergence and Conservatism in Climatic Niches of the Temperate Maples (Acer L.)"

_forests, doi:10.3390/f12050535_

Round 1

Reviewer 1 Report

Comments for Grossman 2021 Forests

Grossman surveyed current distribution of maple species and the associated climatic factors (mean T, seasonality of T, precipitation, and seasonality of the precipitation), and compared similarity or divergence of ecological niches of 184 pairs, some of which are intercontinental, while others are within North America, Eastern Asia, and Eurasia. The paper is well-written and may stimulate further studies on the topic when more samples are included in a similar investigation. Here are my specific comments on the manuscript:

  1. Constrained divergence may require a more explicit definition. Does it imply a spatial pattern of climatic niches? For example, does it mean that in eastern Asia the wide range of temperature and precipitation indicate a less or more constraint on the possibilities of divergence?
  2. Niche divergence and constrained niche diversity seem to be very similar.
  3. Even phylogenies previously published included few subtropical and tropical taxa in Acer, the geographic study here should include them because the complete information reflects the entire range of maple distribution and niches and this study is about niche diversity and differentiation of the taxon. By the way, the author did try to justify the omission of those taxa.
  4. Studies have shown that plastid genes do not reflect phylogenetic relationships in maple and many other plant genera, and the combined analysis by Areces-Berazain et al. did not address the issue, thus, the interspecific relationships may be misleading. So, may be another analysis should be done to include just nuclear genes (ITS included).
  5. Is there a justification for using the 2.5o resolution (111 Km) given that species can be clearly separated within the limit?
  6. It is well known that species of Acer occur in a locality (same LONG x LAT coordinate) but different elevations, and it might be appropriate to include such information if at all possible.
  7. Is the median value the niche optimum? I would image that the niche where most populations occur is the optimal one.
  8. How do you define “dry” and “very dry” environments?
  9. The first sentence in the Discussion section mentioned the “integration of paleontological … data,” but I failed to see any connection with the data in the manuscript.
  10. In 4.1. the statement here “Notably, maple floras on each continent occupy very similar … (Table 1)” seems to be in conflict with Table 1 where Eurasian species appear to be more drought-tolerant with mean annual precipitation ranging from 219 to 907. Also, this differs from the statement in the Abstract “Eurasian species occupy drier habitats than in the bioregions,”
  11. Page 15. “No pair of American taxa hailing from … Table S4)” should Pidentity be < 0.01 instead of >0.05? Less shared niche means small Pidentity.

Author Response

I appreciate this reviewer's thoughtful comments and have improved the original manuscript considerably in responding to their suggestions.

  1. Constrained divergence may require a more explicit definition. Does it imply a spatial pattern of climatic niches? For example, does it mean that in eastern Asia the wide range of temperature and precipitation indicate a less or more constraint on the possibilities of divergence?
  2. Niche divergence and constrained niche diversity seem to be very similar.

In response to this reviewer’s first and second points, I appreciate the observation that constrained niche divergence was not developed appropriately as a concept in the first manuscript draft and needed to be differentiated from constraints per se. I have explicitly developed the term and referenced its earlier use as follows in the Discussion: “I borrow Yang and colleague’s [89] concept of “constrained divergence” to describe this pattern: in which trait divergence is limited by fundamental constraints to adaptive evolution. In the context of Pyron and colleague’s framework, constrained divergence occurs when fundamental limitations on adaptation interact with divergent evolution in response to novel or changing climatic conditions.”

3. Even phylogenies previously published included few subtropical and tropical taxa in Acer, the geographic study here should include them because the complete information reflects the entire range of maple distribution and niches and this study is about niche diversity and differentiation of the taxon. By the way, the author did try to justify the omission of those taxa.

I excluded these taxa primarily out of a desire to focus on niche comparisons within a relatively confined range of temperate climatic conditions. In assessing A. caudatifolium for inclusion in this study, for instance, this species’ precipitation niche was orders of magnitude higher than that of many other study taxa. Furthermore, GBIF coverage for most subtropical taxa was poor. This is now noted in Appendix A: “These taxa were excluded not only to retain a focus on temperate taxa, but also because representative geographical data across their ranges were unavailable for many of them (A. rubescens, A. calcaratum, A. olivaceum, A. elegantulum, A. pauciflorum, A. pubipalmatum, and A. pubinerve).”

4. Studies have shown that plastid genes do not reflect phylogenetic relationships in maple and many other plant genera, and the combined analysis by Areces-Berazain et al. did not address the issue, thus, the interspecific relationships may be misleading. So, may be another analysis should be done to include just nuclear genes (ITS included).

An earlier version of this manuscript entailed phylogenetic analysis using Li and colleagues’ nuclear phylogeny of Acer and generated qualitatively similar results. This is now noted in a new passage in the Methods section: “Use of Areces-Berazain and colleagues’ [17] phylogeny, which integrates signal from both nuclear and plastome sequences, provided qualitatively similar results to analyses carried out with Li and colleague’s [16] nuclear phylogeny (data not shown). As reviewed by both groups, these phylogenies generate trees with similar topologies to each other and to Acer phylogenies published by other workers.” Furthermore, recent exploration of plastid sequence data has, in fact, illuminated phylogenetic relationships in this genus. Representative analyses include Wang et al. (2020; Plant Systematics and Evolution 306:61) and Areces-Berazain et al. (2020; PeerJ 8:e9483).

5. Is there a justification for using the 2.5o resolution (111 Km) given that species can be clearly separated within the limit?

I have now justified this choice in the Methods section: “Climatic data were not assessed at a finer spatial grain in order to provide niche estimates of sufficient generality to compare among taxa across a global scale.”

6. It is well known that species of Acer occur in a locality (same LONG x LAT coordinate) but different elevations, and it might be appropriate to include such information if at all possible.

I added the following rationale for excluding elevation from this analysis: “Though elevation is known to be an important factor in determining species’ distributions [60], it was excluded from this study due to its frequent correlation with temperature and precipitation variables.”

7. Is the median value the niche optimum? I would image that the niche where most populations occur is the optimal one.

The reviewer’s definition of niche optimum makes sense when presence data can accurately reflect the density of individuals of a taxon in a given spot. But without a systematic sampling procedure, the number of points in a given area does not reflect actual abundance. Because presences are thinned, giving an (artificial, but less biased) coverage of points across the species range, median values are suitable for use as niche optima.

8. How do you define “dry” and “very dry” environments?

I very much appreciate this point and reflect on the fact that it is not that Eurasian taxa occupy drier habitats, but rather than East Asian taxa, for instance, occupy much wetter ones. I have adjusted the text to reflect this and avoid the dry/very dry issue the reviewer raised:

“In particular, East Asian taxa grow in a wide range of wet and dry environments, while Eurasian taxa, by comparison, rarely occupy wet habitats (> 1000 mm of precipitation a year) with North American taxa intermediate.”

9. The first sentence in the Discussion section mentioned the “integration of paleontological … data,” but I failed to see any connection with the data in the manuscript.

I agree and have removed this language.

10. In 4.1. the statement here “Notably, maple floras on each continent occupy very similar … (Table 1)” seems to be in conflict with Table 1 where Eurasian species appear to be more drought-tolerant with mean annual precipitation ranging from 219 to 907. Also, this differs from the statement in the Abstract “Eurasian species occupy drier habitats than in the bioregions,”

There are two separate issues here. The passage the reviewer draws attention to refers to temperature niche, not overall niche, in which case the comment that continental floras occupy similar (temperature) niches is in agreement with the table. But, as noted in my response to this reviewer’s comment 8 above, their comments have helped me to more accurately portray the findings regarding dry/wet niches. I have adjusted the Abstract text to read: “Maxent models define a fundamental climatic niche for temperate maples and suggest that drought-intolerant taxa have been lost from the Eurasian maple flora, with little continental difference in temperature optima or breadth.”

11. Page 15. “No pair of American taxa hailing from … Table S4)” should Pidentity be < 0.01 instead of >0.05? Less shared niche means small Pidentity.

I appreciate the reviewer catching this and have revised accordingly.

Reviewer 2 Report

Personally, I do not like MaxEnt as a SDM, due to the bias and "background absences". However, the author has done a great job explaining each part of the process.

Some work should be done with the Results sections (e.g. figure 3 and explanation are not giving valuable information as range of values in niche overlap by node age are large).

Author Response

  1. Personally, I do not like MaxEnt as a SDM, due to the bias and "background absences". However, the author has done a great job explaining each part of the process.

Yes, this is a limitation of the MaxEnt procedure. But true absence data is only available for a handful of maple species, and probably for none outside of North America and western/central Europe. So a synoptic analysis has to rely on pseudoabsences.

  1. Some work should be done with the Results sections (e.g. figure 3 and explanation are not giving valuable information as range of values in niche overlap by node age are large).

Regarding Figure 3 specifically, it is true that node age has low explanatory power as a predictor of overlap. Yet, despite the large spatial and temporal scale of the analysis, this relationship is highly significant. I now give the R^2 for this relationship in the Figure 3 legend and specify in the text that this relationship is weak/modest.

Reviewer 3 Report

- The article is excellent, well designed and conducted.
- It is interesting for specific readers interested on maples biogeography (global o regional concerns).
- The article is difficult to follow in some sections due to the highly detailed information, but it will be welcome to specific readers. In case of readers interested only on a specific group or species, they can go directly to the relevant information.
- Although there are some comments about endangered species of Acer in the conclusions section, it could be welcome some details in the introduction. Many of the Acer species, due to their restricted area (endemicity) are endangered.
- The methodology is well explained and the most critic concerns well justified. Only the selection of 4 global variables of GlobalBioclimatics needs additional explanations. Does the author test other variables? What is the reason why he did not select other bioclimatic parameters such as Bio3 or Bio7 instead of basic climatic values? It is well known that bioclimatics parameters run better in Ecological Niche Modeling than basic climatic values such as bio1 and bio12.
Specific comments:
- A colon is missing at the end of page 4.
- Figure 1 is difficult to read due to their small font size. Alternative: landscape in full page.
- A comma is missing after …(AUC), ….in Figure 2 caption.
- Page 13. First paragraph of Discussion section: …the most synoptic accounting to date of the of the ecological niche…..: of the is repeated.

Author Response

I appreciate this author's care in reviewing my manuscript and have improved it through consideration of their comments:

  1. The article is difficult to follow in some sections due to the highly detailed information, but it will be welcome to specific readers. In case of readers interested only on a specific group or species, they can go directly to the relevant information.

I appreciate this reviewer’s perspective and hope that changes to revised version will allow readers who desire a more synoptic perspective to make it through this manuscript more easily.

  1. Although there are some comments about endangered species of Acer in the conclusions section, it could be welcome some details in the introduction. Many of the Acer species, due to their restricted area (endemicity) are endangered.

I have added the following passage to the Introduction: “Beyond providing a synoptic accounting of the climatic niches of maple taxa for which sufficient phylogenetic and distribution data are available, this approach may also shed light on the climatic forces affecting the distribution of rare and understudied taxa, including the 35 threatened maple species not included in the present study [2].”

  1. The methodology is well explained and the most critic concerns well justified. Only the selection of 4 global variables of GlobalBioclimatics needs additional explanations. Does the author test other variables? What is the reason why he did not select other bioclimatic parameters such as Bio3 or Bio7 instead of basic climatic values? It is well known that bioclimatics parameters run better in Ecological Niche Modeling than basic climatic values such as bio1 and bio12.

I have added the following justification to the Methods section: “Other bioclimatic variables pertaining to temperature or precipitation (e.g., isothermality, bio3) could have been included, but these were largely correlated with the chosen variables across the study system and are less obviously interpretable to lay audiences and managers.” I am not familiar with the common knowledge that derived bioclimatic variables function better as predictors of species’ distributions than basic ones.

  1. A colon is missing at the end of page 4.

Thanks to the reviewer for catching this! I have added the missing colon.

  1. Figure 1 is difficult to read due to their small font size. Alternative: landscape in full page.

I will work with the production team to make sure that Figure 1 is fully readable.

  1. A comma is missing after …(AUC), ….in Figure 2 caption.

I disagree, as the form of the sentence is “X and Y are provided”, no comma is necessary after “Y” (AUC).

  1. Page 13. First paragraph of Discussion section: …the most synoptic accounting to date of the of the ecological niche…..: of the is repeated.

I appreciate the reviewer catching this and have removed the repeating element.

Reviewer 4 Report

Dear Jake J. Grossman,

I have reviewed your manuscript “Evidence of constrained divergence and conservatism in climatic niches of the temperate maples (Acer L.)” submitted to Forests.

The manuscript is well-written, with interesting results, and fits well within the aims and scope of the journal. However, I found it quite long and unbalanced, especially the introduction, where the topic of environmental niche modeling is completely overlooked.  

I have some concerns regarding the methods. By removing records from outside the species native distributions you did not consider what has become known as ‘the equilibrium assumption’, i.e. species are known to be able to tolerate climate conditions beyond those characteristic of their native ranges (see Simberloff 2000; Beaumont et al. 2009; Booth et al. 2014; Booth 2017). Therefore, including records outside the native geographic range is particularly useful to assess species’ tolerance and expand the realized niche, which generally refers to the breadth of conditions under which the species is currently growing.

The use of a fixed set of climate variables for model development based on studies showing these variables constrain plant distribution across spatial scales. Although this approach is not incorrect, it is not the best. Species differ greatly in the way they respond to climate, and therefore, a better approach would have been to identify which variables have a greater effect on individual species; this would have improved the models greatly. This is not mentioned anywhere in the manuscript and should be included as a caveat, especially because of the high number of species. Also, it is mentioned the set of variables used in the study is generally not collinear (L280), yet, there’s no multicollinearity analysis in the study. This analysis should be conducted to validate this statement.

Because there’s no mention of this, I am assuming Maxent's default settings were used for modeling development. This point is very important since the results might have based on this configuration without trying to optimize the model (?). Many articles have explored this problem (see Anderson and Gonzalez 2012; Warren and Seifert 2011; Merow et al. 2013; Radosavljevic and Anderson 2013; Syfert et al. 2013; Halvorsen et al. 2016). I would have expected that at the very least, and to increase the quality of the models, the ‘hinge’ and ‘threshold’ features would have been deactivated to avoid overfitting response curves.

Regarding the evaluation of model performance, it is only used the AUC value, which does not show much, only that the model is not bad. Getting high values of AUC is very common for models based on presence-only data such as the one developed in this work. If the model is evaluated with cross-validation (for example, using spatial blocks, as in Roberts et al. 2017, http://onlinelibrary.wiley.com/doi/10.1111/ecog.02881/full), a high value of AUC would be more convincing. AUC levels for good performance are only relevant for presence/absence data. Furthermore, problems with AUC are well documented (see Lobo et al. 2007). Another option is to calculate an additional statistic, such as ‘true skill statistics (TSS)’, to corroborate model performance. This is especially important because the AUC values in this work were quite low (mean 0.720), reflecting not quite good model performance. Yet, there’s no mention of this in the discussion.

I recommend including in the discussion a section of caveats and limitations. Finally, the conclusion has topics that should be moved into the discussion.

See more specific comments in the file attached.

Author Response

This reviewer has pointed out several key shortcomings in the manuscript as originally submitted and I believe that their suggestions have much improved this revised version.

  1. The manuscript is well-written, with interesting results, and fits well within the aims and scope of the journal. However, I found it quite long and unbalanced, especially the introduction, where the topic of environmental niche modeling is completely overlooked. 

I appreciate that this reviewer found the manuscript well-written and topical. Their observation that SDM was not really addressed in the Introduction makes sense. I struggled with this because, though Maxent SDM modeling is a/the central tool used in this manuscript, the output of the models themselves (as opposed to niche comparison tests) are not the focus of the results. To help preview the Maxent approach in the Introduction, I moved the following text from the Methods to the Introduction itself: “Given the availability of presence data but not true absence data for the study taxa, I elected to use the Maximum Entropy or Maxent modeling [54,55] to build ENMs for each taxon. Maxent is widely used in ecological modeling; though it is vulnerable to sampling bias, misspecification of environmental background, inclusion of correlated predictor features, and other pitfalls of ENM, these potential problems can be addressed through appropriate model specification [56]. Briefly, Maxent models integrate presence data for a taxon of interest and user-provided environmental data for each point throughout a user-defined spatial extent to produce estimates of the likelihood of the taxon’s occurrence across that extent.”

  1. I have some concerns regarding the methods. By removing records from outside the species native distributions you did not consider what has become known as ‘the equilibrium assumption’, i.e. species are known to be able to tolerate climate conditions beyond those characteristic of their native ranges (see Simberloff 2000; Beaumont et al. 2009; Booth et al. 2014; Booth 2017). Therefore, including records outside the native geographic range is particularly useful to assess species’ tolerance and expand the realized niche, which generally refers to the breadth of conditions under which the species is currently growing.

Again, this reviewer has focused on a critical challenge that I have dealt with in deciding how to do this analysis. My choice to generally excluded occurrences outside of the non-native range except for four widely naturalized species does indeed fail to take into account the fact that species can be cultivated or occur spontaneously outside of their native ranges. Yet in doing so, I also sought to strike a balance between this issue and the insight that many occurrence points outside of native ranges (especially in GBIF data) reflect occurrences in which a plant has been protected from environmental extremes (through watering, nursery propagation through the seedling phase, and through the urban heat island effect) during cultivation in a private or public garden. In fact, a previous version of this manuscript was rejected from a different journal because a reviewer objected to the inclusion of any occurrence points outside of species’ native ranges. Because of this, I prefer to stick to my current conservative approach and have engaged with this reviewer’s concerns in the new Limitations section.

  1. The use of a fixed set of climate variables for model development based on studies showing these variables constrain plant distribution across spatial scales. Although this approach is not incorrect, it is not the best. Species differ greatly in the way they respond to climate, and therefore, a better approach would have been to identify which variables have a greater effect on individual species; this would have improved the models greatly. This is not mentioned anywhere in the manuscript and should be included as a caveat, especially because of the high number of species. Also, it is mentioned the set of variables used in the study is generally not collinear (L280), yet, there’s no multicollinearity analysis in the study. This analysis should be conducted to validate this statement.

Yes, I agree that my approach is far from perfect. Indeed, there is even a great deal of variability in the extent to which each of the four chosen variables influence species’ distributions. Caveats about the limitations of using a fixed set of climatic variables to predict diverse species’ distributions are now included in the limitations section.

I now include the results of collinearity analysis in Table S3 and in the main text.

  1. Because there’s no mention of this, I am assuming Maxent's default settings were used for modeling development. This point is very important since the results might have based on this configuration without trying to optimize the model (?). Many articles have explored this problem (see Anderson and Gonzalez 2012; Warren and Seifert 2011; Merow et al. 2013; Radosavljevic and Anderson 2013; Syfert et al. 2013; Halvorsen et al. 2016). I would have expected that at the very least, and to increase the quality of the models, the ‘hinge’ and ‘threshold’ features would have been deactivated to avoid overfitting response curves.

Again, these points are well-taken. I did choose to use default settings, in part because these are the settings employed when niche comparisons are made in the ENMTools procedure for using Maxent models to perform identity and background tests. This may have indeed produced suboptimal models. To account for this deficiency, I have now altered the Methods to specify that default options were used and reflect further on the issue this reviewer has raised in the Limitations section.

  1. Regarding the evaluation of model performance, it is only used the AUC value, which does not show much, only that the model is not bad. Getting high values of AUC is very common for models based on presence-only data such as the one developed in this work. If the model is evaluated with cross-validation (for example, using spatial blocks, as in Roberts et al. 2017, http://onlinelibrary.wiley.com/doi/10.1111/ecog.02881/full), a high value of AUC would be more convincing. AUC levels for good performance are only relevant for presence/absence data. Furthermore, problems with AUC are well documented (see Lobo et al. 2007). Another option is to calculate an additional statistic, such as ‘true skill statistics (TSS)’, to corroborate model performance. This is especially important because the AUC values in this work were quite low (mean 0.720), reflecting not quite good model performance. Yet, there’s no mention of this in the discussion.

Following this reviewer’s suggestion, I have calculated TSS for all models and report these values in Table S3. I have also added the following text to the Methods section: “As an alternative, I also calculated and report the True Skills Statistic (TSS; TPR + True Negative Rate [TNR] – 1] alongside the training AUC. TSS ranges from -1 to 1, with values above zero indicating better model performance by chance.”

  1. I recommend including in the discussion a section of caveats and limitations. Finally, the conclusion has topics that should be moved into the discussion.

I appreciate this constructive suggestion and have split up the current Conclusion. The existing discussion of limitations in the former Conclusion section are now, along with caveats as suggested by this reviewer, unpacked in a Limitations section of its own. Other text has been integrated back into the Discussion.

  1. See more specific comments in the file attached.

I appreciate this reviewer’s attentive reading of the manuscript. I have responded to all of their suggestions in comments in the attached PDF.

Round 2

Reviewer 4 Report

Dear Jake J. Grossman,

Thank you for addressing all my comments and concerns. I particularly appreciate the inclusion of the "Limitations" section. The manuscript will be a great contribution to the field. 

I just have one minor comment. L725-727, are these sentences meant to be in brackets? It doesn't seem right.